# Variational AutoEncoder to Identify Anomalous Data in Robots

**Luigi Pangione \***, **Guy Burroughes** and **Robert Skilton**

Remote Applications in Challenging Environments (RACE), United Kingdom Atomic Energy Authority, Abingdon OX14 3DB, UK; guy.burroughes@ukaea.uk (G.B.); robert.skilton@ukaea.uk (R.S.)
\* Correspondence: luigi.pangione@ukaea.uk

**Abstract:** For robotic systems involved in challenging environments, it is crucial to be able to identify faults as early as possible. In challenging environments, it is not always possible to explore all of the fault space, thus anomalous data can act as a broader surrogate, where an anomaly may represent a fault or a predecessor to a fault. This paper proposes a method for identifying anomalous data from a robot, whilst using minimal nominal data for training. A Monte Carlo ensemble sampled Variational AutoEncoder was utilised to determine nominal and anomalous data through reconstructing live data. This was tested on simulated anomalies of real data, demonstrating that the technique is capable of reliably identifying an anomaly without any previous knowledge of the system. With the proposed system, we obtained an F1-score of 0.85 through testing.

**Keywords:** condition monitoring; robot; VAE; anomaly detection

## 1. Introduction

With robotic systems involved in nuclear operations, it is crucial to be able to identify anomalies as early as possible. In radiation environments, where human access is not possible, being able to identify a problem in early stages can allow the operator to pause operations and relocate the robot to a place where it is possible to perform necessary maintenance. Moreover, in such environments, robots suffer from early ageing due to the radiation dose they are exposed to. The effects of radiation can induce the gradual degradation of robot performance as well as sudden failures. Radiation can have diverse effects on a range of components of the robot, including those that would be considered robust in normal operations. It is clear then, that in a radiation environment, the appearance of a fault may be a dramatic event. A robot unable to move can have a dramatic impact on safety. Moreover, such conditions can have a serious impact on operational costs as it may be highly difficult to recover the robot for repair. It is worth noting, in fact, that robotic systems for nuclear operations often require bespoke solutions with few alternatives.

A good example of a challenging environment is that of nuclear gloveboxes. These provide the operator with a very limited workspace, prone to clutter, with a vision from the outside not always optimal. Moreover, an operator is equipped with personal protective equipment such as coveralls and masks which reduce the ability to move and see. Furthermore, the processed object can contain hazardous materials that are difficult to assess under such conditions. In typical glovebox operations, objects that need processing are inserted inside the glovebox through a sealed door; once the objects are secured inside the glovebox, the operator executes all the required tasks; at the end, the processed objects are posted out and the glovebox is prepared for the subsequent task. In this work sequence, it is extremely important that the operator can complete all the tasks assigned to it without any interruptions. It is clear then that, in a robotic glovebox, information on the status of the robot and its ability to complete the tasks without any faults occurring is of paramount importance.

The purpose of a condition monitoring system (CMS) is to monitor measurements taken from the robotic system, and infer the health of the device, including the possibility

of unusual behaviour or the degradation of performance, and report these findings to the robot's operator. Traditional CMS uses dedicated sensors to identify a fault in the components; for example, a vibration sensor to identify the faults on a motor. In our work, we made use of already existing data provided by the robot hardware to the operator and to the control system. As will subsequently become clearer, an anomaly in these data is not necessarily related to a fault in a robot component, but represents an unexpected event in a wider sense. Differences in measurements of parameters such as position and velocity can be noted by an expert operator without any additional system. Other measurements, such as those of motor current, torque and temperature, are usually hidden to the operator to avoid distractions. Moreover, the wealth of information available to operators during a complex robot operation may be overwhelming. Variations in such measurements are therefore impossible to be noted by an operator, even the most experienced one. From the operator's point of view, it is important to remain focused on performing the task and be able to be informed with only the most relevant information in case a fault is developing.

In our work, we used Variational AutoEncoder to identify anomalies in our robotic glovebox setup. This choice was motivated by the highly complex and structured nature of the relationship between the measured signals and the robot's health. Our main research contribution is to propose a Monte Carlo-based technique to produce a statistic of expected nominal behaviour and then use this to identify anomalies. To improve our results, we also scored samples by using loss function scoring and we made use of the F1 score and ROC score to tune the sensitivity to discriminate anomalies.

This paper is organized as follows. In the next section, we give a background on anomaly detection and Variational AutoEncoder. In Section 3, we introduce a technique to use the anomaly detection. In the subsequent section (Section 4), we introduce our experimental setup. In Section 5, we report and discuss our results and in the final section, we discuss our conclusions and outline future works.

## 2. Background

### 2.1. Anomaly Detection

Traditional fault detection techniques require detailed a priori knowledge of all the possible faults that a robot may encounter. However, this is not always possible in challenging environments, as access to extensive characterisation is rarely feasible. This leads to many faults occurring in a nuclear environment (for example) that are novel. However, the existence of a fault can be inferred by a discrepancy with respect to the usual behaviour in the robot's data. Such discrepancy, or anomaly, in the data can represent different types of data anomaly. In [1], the authors classify anomalies in the following three categories:

- Point anomalies—where a single instance of the data is anomalous with respect to all the rest of the data;
- Contextual anomalies—where an instance of the data is anomalous with respect to the specific context of the data; i.e., data that would be nominal in the context of a robot's linear motion would also be anomalous when the robot performs accelerating motion;
- Collective anomalies—where a collection of the data is anomalous with respect to the data set; e.g., data from the current sensor and thermometer are individually nominal, but not both at the same time.

### 2.2. Variational AutoEncoder

In recent years, deep learning-based generative models have gained more and more interest due to (and implying) some amazing improvements in the field. One such technique is the Variational AutoEncoder (VAE) [2]. In probability model terms, the VAE refers to approximate inference in a latent Gaussian model where the approximate posterior and model likelihood are parametrised by neural networks (the inference and generative networks). In neural network language, a VAE consists of an encoder, a decoder, and a loss function.

The purpose of the encoder is to map the information included in the sample in a reduced dimension space, called latent space. This space is meant to contain the main characteristics of the samples. The decoder, on the other hand, maps a sample from its latent space representation back to the original form. The peculiarity of the VAE is that each dimension of the latent space consists of a Gaussian distribution, each of which is characterised by a mean and a logarithmic variance value. This implies that, once a sample is mapped in the latent space, it is possible to draw multiple times to obtain multiple reconstructions of the original sample. In a VAE, the loss function is the sum of two parts: reconstruction loss and latent loss. The reconstruction loss is a metric of the VAE ability to reproduce the desired output; for example, such a loss can be the mean square error (MSE) or the mean absolute percentage error (MAPE). The latent loss encourages the latent space to have a form of Gaussian distribution; an example of latent loss is the KL divergence loss.

In recent years, VAE have been used for anomaly or fault detection in a wide range of applications, from images to bank transactions. In [3], the authors combined VAE and long short-term memory (LSTM) to detect anomalies in time series. In [4], the authors used the VAE model to detect anomalies in videos. It is interesting to note that in the paper, the latent space is modelled as a Gaussian mixture model (GMM) rather than a single Gaussian distribution. In [5], the authors took advantage of multiple draws from the latent space to map the reconstruction error, i.e., the difference between the input sample and its reconstruction into Gaussian distribution. We do not think it is possible to apply the same techniques to our data and therefore, as it will be clearer later in the paper, we adopted a different method to identify anomalies. In [6], the authors discriminated anomalies by clustering the latent space. Furthermore, in this case, we do not believe it is possible to apply this technique to our data. In [7], the authors explored the use of conditional VAE for anomaly detection in CERN LHC data.

## 3. VAE for Anomaly Recognition

### 3.1. Reconstruction

We trained the VAE to reproduce in the output the sample presented in the input. The main idea was that the VAE would be able to reproduce a sample that already appeared during training, while it would fail if a sample contained any kind of anomaly. A sample is made by measurements collected from all the joints; in the case of an anomaly in a joint, only some of the measurements will be affected.

Figure 1a,b illustrates a simplification of the reconstruction concept. In particular, in normal conditions represented by Figure 1a, measurements collected from the robot are collectively known, and therefore the VAE is able to reproduce all of them correctly. In the case of measurements not already being collectively presented during training, as shown in Figure 1b, the VAE will not be able to correctly reproduce them.

It is important to note that in Figure 1b, the anomalous measurement must not necessarily be novel or contain values never seen before. The VAE will not be able to reproduce all of these as long as they are not collectively the same.

One way of seeing this is that the state of the machine is then encoded in the latent space. If the encoder encodes a region of the latent space that has not been trained, the decoder will not be able to decode and thus coherently/correctly reproduce the values. Following this analogy, using a VAE allows the system to account for sensor noise, and the latent space can encode a covariance to the probability of values based on the region.

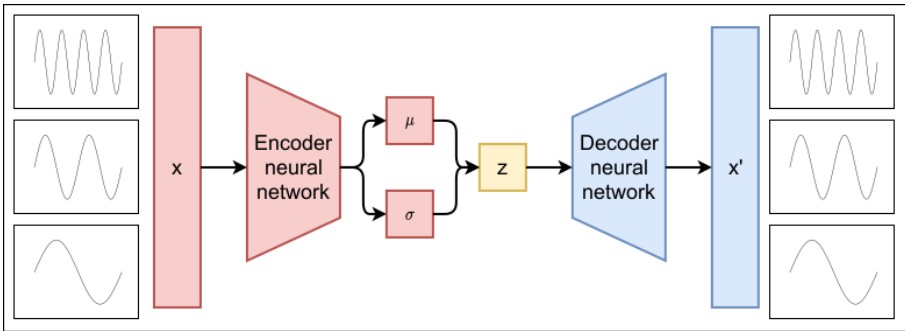

(**a**) Normal condition behaviour.

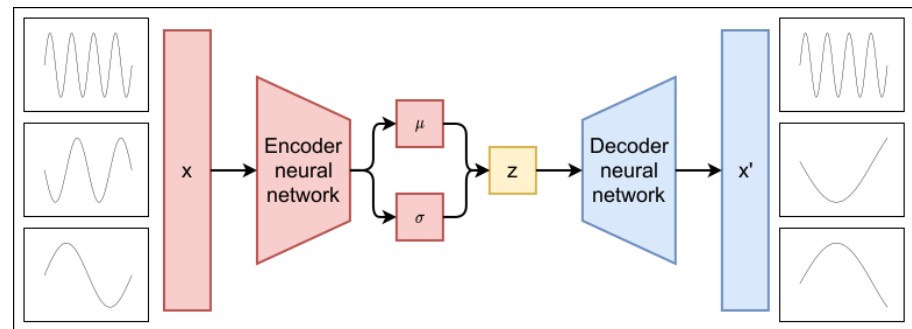

(**b**) Anomalous condition behaviour.

**Figure 1.** Simplified scheme of VAE reproducing data under normal conditions with anomalies.

### 3.2. Monte Carlo Reconstruction

One of the main advantages of using a VAE instead of an AutoEncoder is that having a stochastic process as part of the latent space permits the generation of multiple reconstructions of the predicted signal, starting from a single point in the latent space in a Monte Carlo fashion. By separating the encoder and the decoder components of a trained VAE, it is possible to use the encoder to obtain a latent space representation of a sample. From here, it is then possible to sample the decoder multiple times, in a Monte Carlo fashion, to collect a statistic of the expected reconstruction behaviour. The reconstruction of any sample which is not compliant with this statistic can be interpreted as an anomaly. Another advantage of using VAE is its robustness against noise. It is inevitable that there will be a base level of noise in any sensor reading, as it is the case in entropy, so it will not be possible for the decoder to reproduce this signal component exactly. However, the level of noise compared to the signal could be encoded into the covariance of the VAE.

It is assumed that the noise can be approximated as a Gaussian in the latent space, as the latent space would approximate the underlying parameters of the system. A Monte Carlo ensemble decoded from the latent space can then approximate a nominal stochastic distribution.

For example, it is possible to generate a zone around each signal showing nominal behaviour. This zone can be calculated as the convex hull of all the expected reconstructed measurements. Each signal reconstructed within this zone can be considered as nominal behaviour, while the reconstructed sample outside of it can be considered an anomaly. A Gaussian mixture model was investigated but deemed unnecessary. A schematic representation is shown in Algorithms 1 and 2. In particular, Algorithm 1 describes the process of generating the zone in which a sample can be considered as having a nominal behaviour, while Algorithm 2 describes the process of assessing the status of a sample.

---

**Algorithm 1:** Algorithm to construct nominal behaviour zone.

```
/* Retrieve encoder and decoder from a trained VAE          */
encoder = trained_vae.get_encoder();
decoder = trained_vae.get_decoder();
/* Iterate over all the samples                             */
for sample s do
    /* get latent space l(s) representation of the sample s  */
    l(s) = encoder(s);
    for draw=1:1e6 do
        /* get reconstructed sample s* from decoder          */
        s* = decoder(l(s));
        /* store all reconstructed sample s*                 */
        set_of_s*.append(s*)
    end
    /* define normal behaviour as envelope of all reconstructed
       samples s*                                            */
    for each signal do
        signal_normal_behaviour(s) = envelope(set_of_s*, signal)
    end
end
```

---

**Algorithm 2:** Algorithm to assess the status of a sample.

```
/* Iterate over all the samples                             */
for sample s do
    /* get reconstructed sample s* from vae                  */
    s* = vae(s);
    /* assess against calculated nominal behaviour zone      */
    for each signal do
        if s* ∈ signal_normal_behaviour(s) then
            s_is_anomaly = False
        else
            s_is_anomaly = True
        end
    end
end
```

## 4. Experimental Setup

### 4.1. Glovebox Use-Case

The setup consists of two Kinova Gen 3, with seven degrees of freedom [8] and two robotic arms, installed in the glovebox demonstrator [9]. The two robotic arms are identical, but the end effectors attached to them considerably differ in terms of dimensions and weights. The experiment will perform a series of CMS dedicated moves that are executed at the beginning and at the end of operations. There are many advantages of using dedicated moves, however, the main one is that they provide a solid and reliable base of data not affected by any external factor, such as human error. Moreover, in a robotic glovebox context and from the operational point of view, it is convenient to have "warm-up" and "cool-down" phases at the beginning and at the end of operations, during which one can ensure that the system is, respectively, ready to operate or has not been damaged during the operations.

The glovebox system is controlled using the ROS framework [10,11]. The Kinova ROS package provides a large amount of real-time data sampled at 1 KHz for each joint. In particular, in our work, we use the joint position, joint velocity, motor current and motor torque of each joint.

It is very important to note that the measurements in use are very different from each other and their range of extension is very diverse. This makes it very difficult to scale them to a common range, i.e., between 0.0 and 1.0. Scaling each individual measurement with a common scaling factor would have made features disappear; on the other hand, scaling them with a measurement's specific factor would have modified the relationship between measurements—unless all data were scaled equally, which would potentially suppress some values.

Not having all the input normalised between 0.0 and 1.0 creates a constraint in the definition of the reconstruction loss. In fact, the same mean square error in two different measurements can have different effects. For this reason, this work evaluates the reconstruction error using the mean absolute percentage error (MAPE). Using MAPE reconstruction loss, the same error in two different measurements is considered differently according to the measurement magnitude.

To be able to capture the dynamic behaviour of the system, we chose to consider, as a single sample, the data collected during a configurable time window. A sample is therefore a matrix where each row represents a measurement and the columns represent the acquisition times. In Figure 2, an example of the data coming from joint 3 sectioned in the time window is presented.

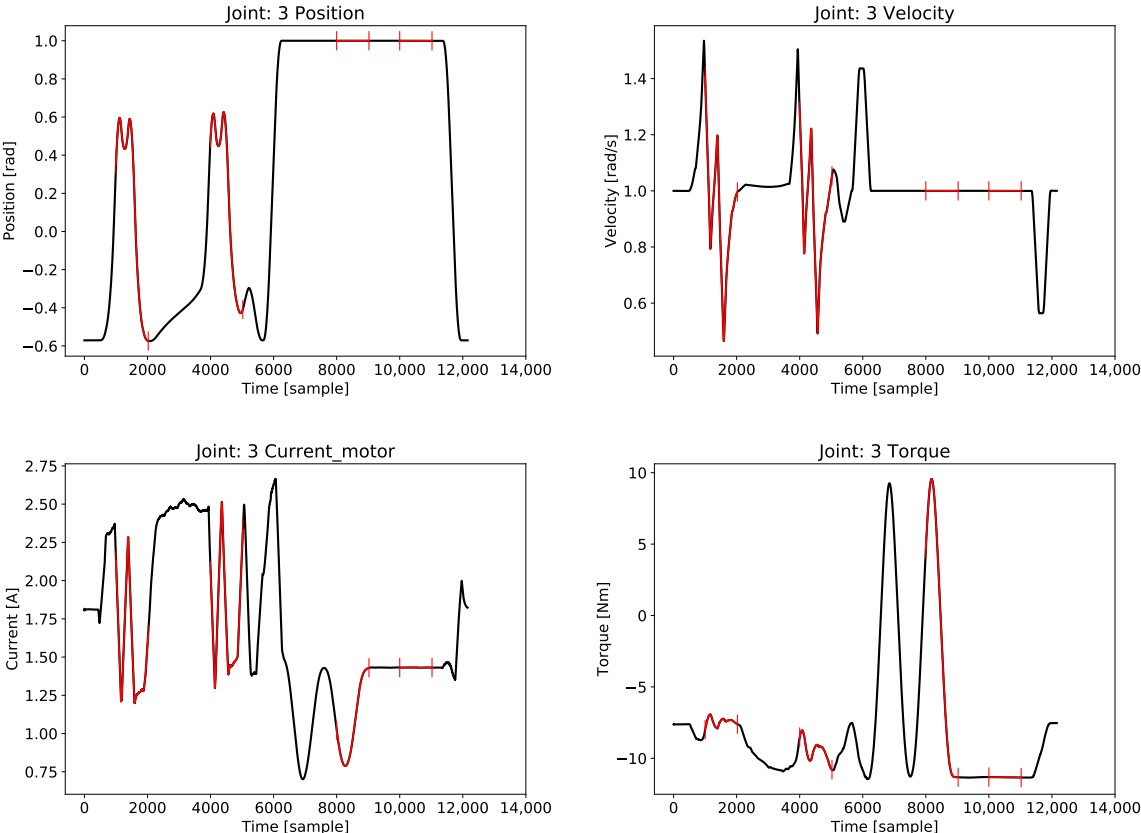

**Figure 2.** An example of time-windowed data sampled from a single joint for one glovebox robot.

It is important to note that this choice does not affect the capability of the system to identify anomalies while working online in any way, as the current sample at time $t_{now}$ should be the set of data collected during the time window $[t_{now} - t_{window}, t_{now})$. It is also important to note that the longer the time window, the highest the amount of information each sample contains is. On the other hand, using a long time window, the system becomes less sensitive to short time perturbations. As will be explained later, the length of the time window changes the behaviour of the system in identifying different types of faults.

### 4.2. Implementation

Our VAE layout consists of a fully connected multiple-layer neural network. Through testing, an optimal number of layers and their sizes were determined. Good results in reproducing the input samples were obtained with the encoder's layers dimensioned $512, 256, 128, 64, 32$ and with a latent space with a dimension of 6. As the sample's values are not bonded between 0.0 and 1.0, a Leaky ReLu activation function was used. The decoder was implemented in a symmetric way. For operational reasons, the VAE was trained using data coming from only one of the two robots installed in the glovebox, from now on the training robot. Data collected from the other robot, now referred to as the testing robot, were used for comparison purposes.

In Figure 3, it is reported that data were reconstructed by the trained VAE. In particular, reconstructed measurements of Joint 3 using data collected from the training robot are reported in the figure, but not those used during VAE training. Black lines represent original measurements, while the red lines represent the reconstructed measurements. To increase the readability of the figure, the reconstructed data were reported only during certain time windows.

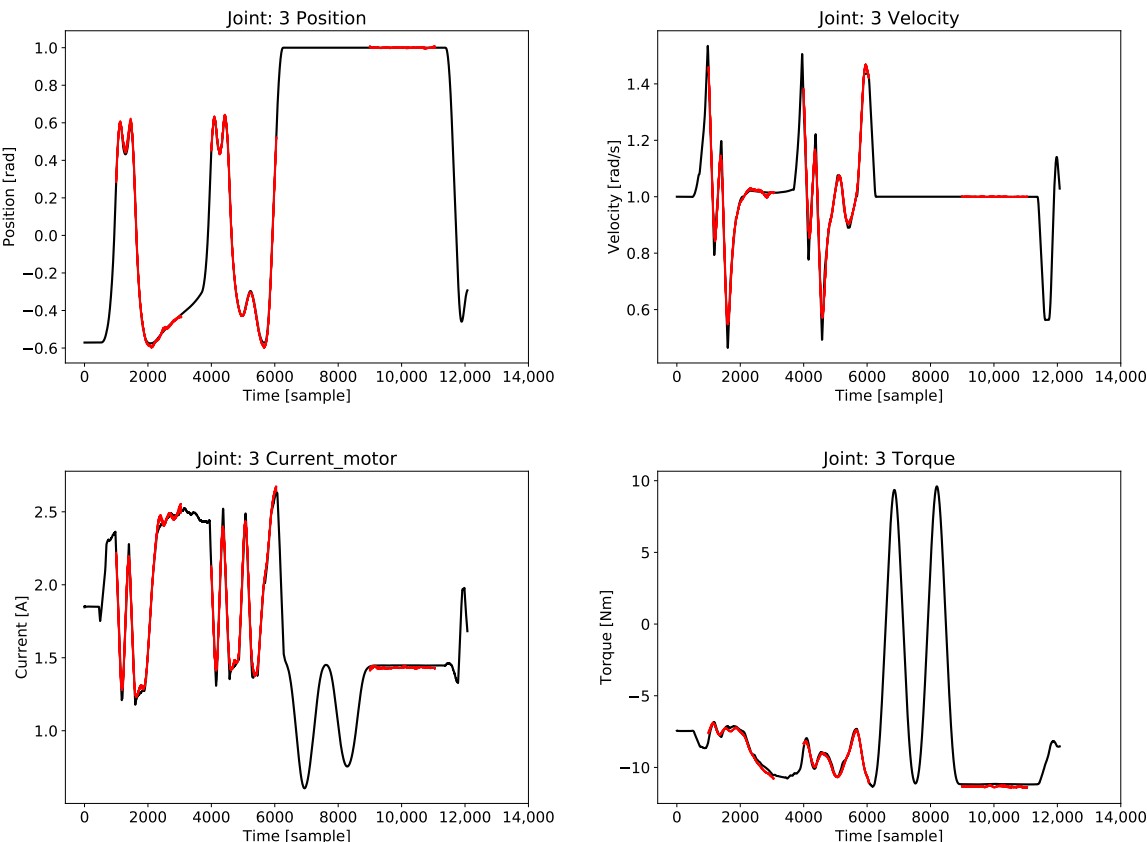

**Figure 3.** Example of data reconstructed by VAE. Original measurements not used during training are reported in black. Different samples of output reproduced by the trained VAE are reported in red.

In Figure 4, we report the mean value of each latent space dimension over time. Red lines represent results obtained from the training data, while blue lines represent the results obtained from the testing robot data set.

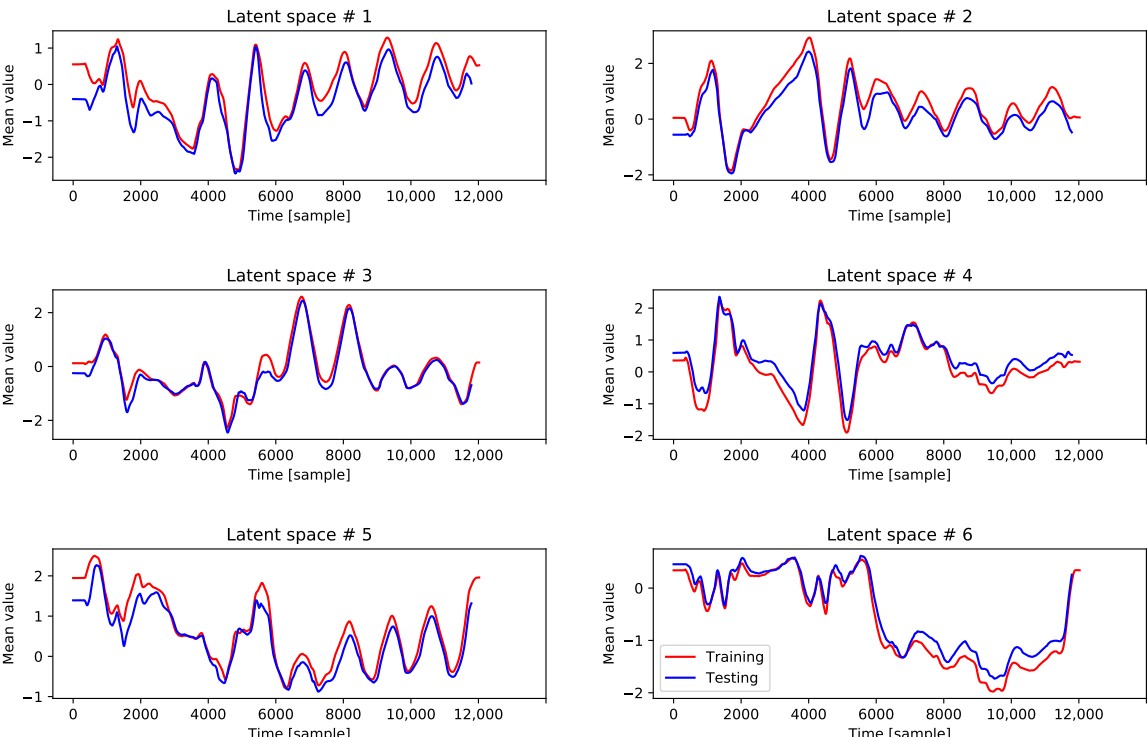

**Figure 4.** Example of data represented in latent space. Red lines represent results obtained from training data, while blue lines represent results obtained from testing data.

## 5. Results

We used the Monte Carlo technique explained in Section 3.2 to generate a zone in which we expected a signal with nominal behaviour.

In Figure 5a, the calculated zone in which the motor current of joint 3 is expected to be reconstructed is shown in red, while the black line represents the actual measure. As can be noted, the zone of expected nominal behaviour is very narrow.

In Figure 5b, multiple draws of samples collected from the testing robot (in blue) are compared to the expected behaviour zone (in red). Data from the testing robot are clearly different from the nominal behaviour and are definitively outside the zone of expected behaviour. This is because the two robots are equipped with two different end effectors.

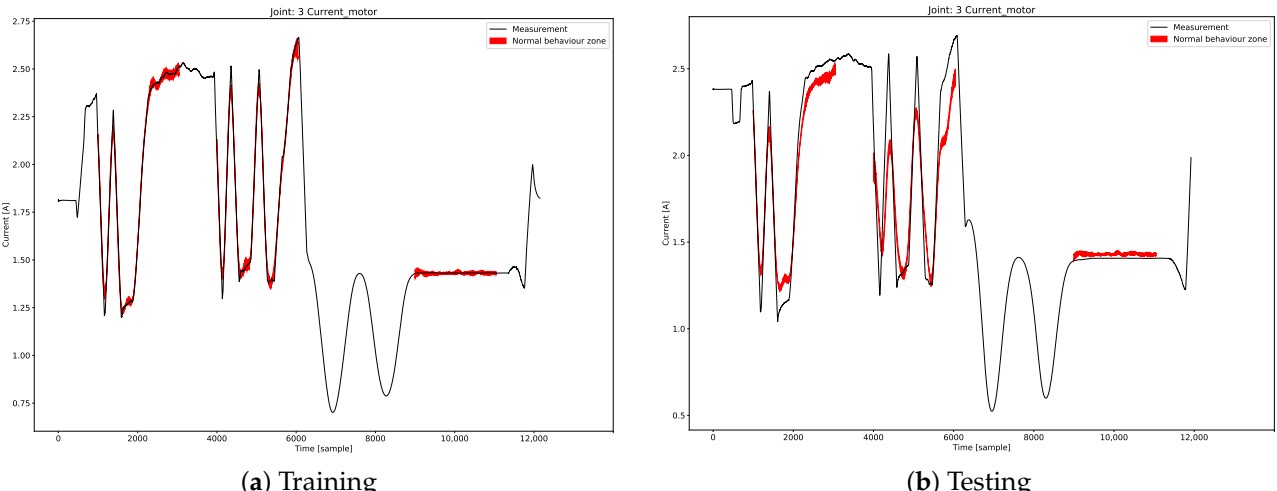

(**a**) Training        (**b**) Testing

**Figure 5.** Example of calculated nominal behaviour zone and its comparison with multiple reconstructions in the case of the training and testing robot data.

To improve the identification of anomalies, we opted to use the VAE loss function to score a sample. We only focused on the training robot because the data coming from the testing robot were already proven to be anomalous. As we did not experience any anomaly with the training robot over time, we modified our data to generate simulated anomalies. In particular, we modified our data in two different ways:

(a) Variation perturbation—a variation of 20% in some of a joint's measurements;
(b) Swap perturbation—a swap between two time windows of some of the joint's measurements.

The first type of data perturbation was intended to reproduce a "point anomaly", where the perturbed data are anomalous with respect to all the rest of the data. In this sense, we can imagine this anomaly as the data that assume values never seen before during the training.

The second type of data perturbation was intended to reproduce a "contextual anomaly", where the perturbed data are anomalous in its context. In this sense, we can imagine this anomaly as data that are not new, but anomalous because they are in the wrong context.

We considered an anomaly as a sample for which the loss score is higher than a predefined threshold. We calculated this threshold as the value that provides the maximum F1 score. It is important to note that this is possible because we generated simulated faults, and therefore, we obtained ground truth information on the data.

In Figure 6a,b, different values of the F1 score and ROC score are reported for different values of the time windows in the case that the data are modified using a "variation perturbation".

Our results show that for this type of anomaly, the best F1 score is approximately 0.80, and it is obtained with a time window of 2 samples (dark blue line). Small time windows (2, 64, 128, 256) do have similar good performances, while longer time windows (1024 and 2048) have the worst performances. These results were also confirmed by the ROC score, in which small time windows' curves are closer to the top left corner. Intuitively, this was equivalent to saying that as point anomalies are data that have never been seen before, they are easier to recognise using short time windows.

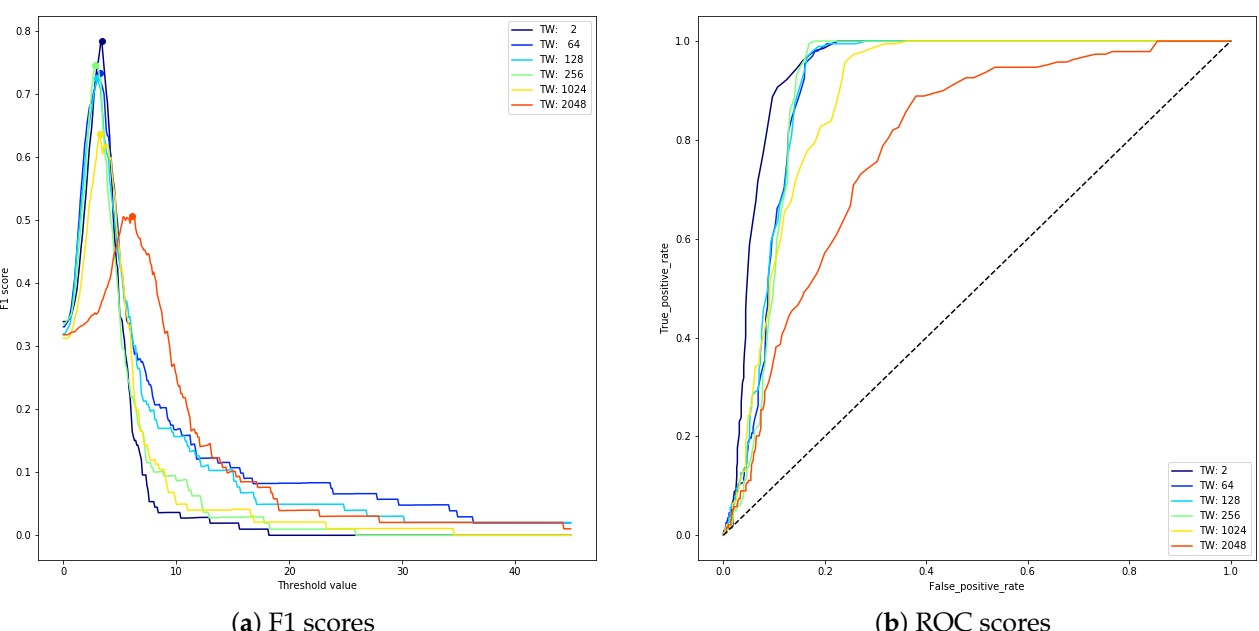

(**a**) F1 scores　　　　　　　　　　　　　　　(**b**) ROC scores

**Figure 6.** Trends of F1 score (**a**) and ROC score (**b**) curves for different values of time windows in the case of simulated variation perturbation anomalies.

Similarly, in Figure 7a,b, different values of the F1 score and ROC score are reported for different values of the time windows in the case that data are modified using a "swap perturbation".

Our results show that for this type of anomaly, the best F1 score is approximately 0.85, and it was obtained with a time window of 1024 (yellow line), which corresponds to approximately 2 s. Furthermore, in this case, it is possible to observe how long time windows and short time windows perform in opposite ways. Intuitively, long time windows perform better with the "swap perturbation" anomaly because the data are not novel in value in this context, therefore the system needs more information to identify the anomalies.

Overall, these results show that the VAE provides very good accuracy in both types of simulated anomalies.

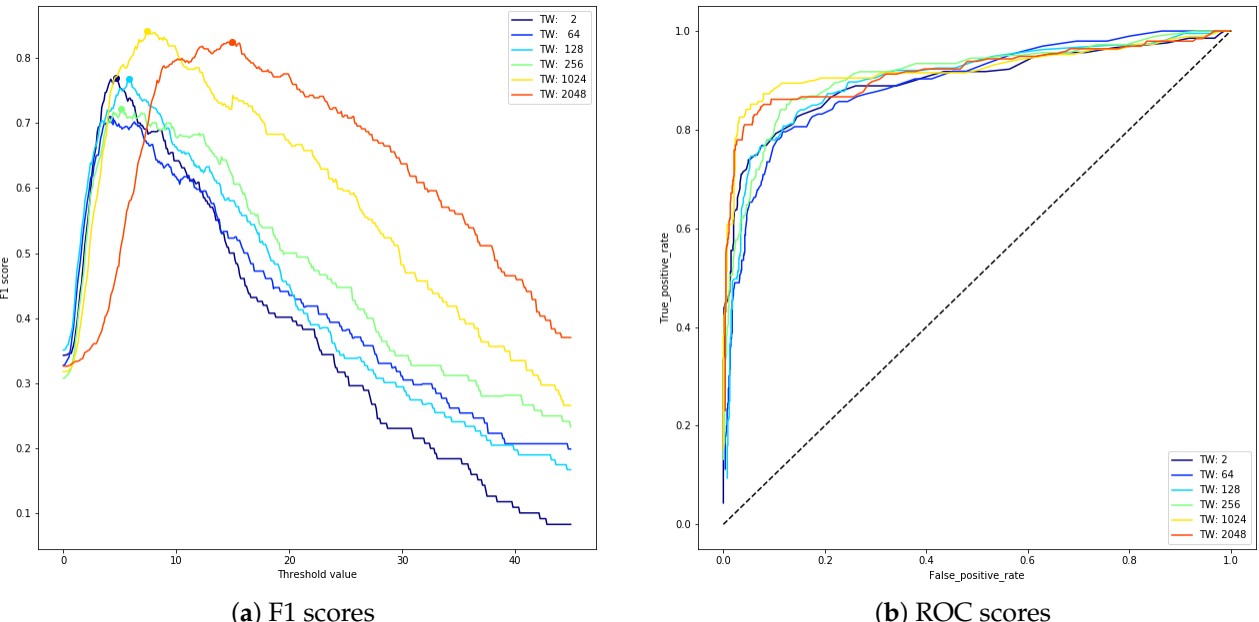

(**a**) F1 scores　　(**b**) ROC scores

**Figure 7.** Trends of F1 score (**a**) and ROC score (**b**) curves for different values of time windows in the case of simulated swap perturbation anomalies.

## 6. Discussion and Conclusions

In this paper, we investigated the use of VAE in identifying anomalous data collected from our robotic glovebox setup. We defined a Monte Carlo-based technique to produce a statistic of expected nominal behaviour results. We applied this technique to data collected from two identical robots equipped with different end effectors. To improve our results, we used a loss function score against simulated anomalies in the data. We proved that both techniques can be used in the detection of anomalies in the data.

The stochastic process in the latent space allowed us to encode a sample into a latent space and then draw from it multiple times, to obtain a statistic of the VAE ability to reconstruct that sample. In contrast to [5], we decided to not assume that the reconstruction error was Gaussian. Instead, we decided to use these statistics differently. Unfortunately, the zone we obtained was too narrow to be practically useful. However, this may not be true for less stable and more fault-prone systems.

As an alternative, we opted to use the VAE score to assess whether the sample was an anomaly. To calculate the threshold to discriminate normal from anomalous behaviour, we made use of F1 and ROC scores. Interestingly, the length of the time window influenced the ability to identify different types of simulated anomalies. In particular, in simulated context anomalies obtained by swapping values of different instants in time for some measurements, long time windows performed better. This was expected as in this type of anomaly, values were presented during training, but not collectively at the same time. This

required more information to be available in the sample. On the other hand, in simulated point anomalies, obtained by increasing the values of some measurements by 20%, the VAE did not need much information to identify values which had never been presented before. In this case, short time windows performed better.

One of the weaknesses of this study is the lack of any real anomaly data vs. actual anomaly data. During our time with the Kinovas, we only witnessed one anomaly in 100 s of hours of operations, which was identified by the proposed system. For future studies, we propose that a more anomaly-prone system be used, rather than an industrial robot, which are known for their robustness.

In future works, we will further investigate the possibility of using statistics from multiple reconstructions of the same sample to identify anomalies or using a standard auto-encoder to simplify the process.

**Author Contributions:** Conceptualization, L.P., G.B. and R.S.; methodology, L.P., G.B. and R.S.; software, L.P.; validation, L.P. and G.B.; formal analysis, L.P., G.B. and R.S.; investigation, L.P. and G.B.; resources, G.B. and R.S.; data curation, L.P.; writing—original draft preparation, L.P.; writing—review and editing, L.P., G.B. and R.S.; visualization, L.P.; supervision, G.B. and R.S.; project administration, G.B. and R.S.; funding acquisition, G.B. and R.S. All authors have read and agreed to the published version of the manuscript.

**Funding:** This project was supported by the RAIN Hub, which was funded by the Industrial Strategy Challenge Fund as part of the government's modern industrial strategy. The fund was delivered by the UK Research and Innovation and managed by EPSRC [EP/R026084/1].

**Institutional Review Board Statement:** Not Applicable.

**Informed Consent Statement:** Not Applicable.

**Data Availability Statement:** Not Applicable.

**Conflicts of Interest:** The authors declare no conflict of interest.

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
