# Peer review of "Variational AutoEncoder to Identify Anomalous Data in Robots"

_robotics, doi:10.3390/robotics10030093_

Round 1
Reviewer 1 Report
- Lack of the code which would allow reproducing the result,
- can you compare your results with similar works,
- better justification of using VAE instead of AE should be provided,
- it would be interesting to visualize how different data is presented in the bottleneck and this could be used as a form of explanation of the result? This is where VAE could be helpful
Author Response
Comment |
Reply |
Lack of the code which would allow reproducing the result |
We agree with the reviewer and in fact we are planning to publish paper’s code on public company’s github account. |
can you compare your results with similar works |
We found no current work that would serve as fair comparison. |
better justification of using VAE instead of AE should be provided |
In subparagraph 3.2 "Monte Carlo Reconstruction" modified sentences into: “One of the main advantages of using a VAE instead of AutoEncoder is that having a stochastic process as part of the latent space permits to generate multiple reconstructions of the predicted signal starting from a single point in the latent space in a Monte Carlo fashion” And “Another advantage of using VAE is its robustness against noise” |
it would be interesting to visualize how different data is presented in the bottleneck and this could be used as a form of explanation of the result? This is where VAE could be helpful |
Added Figure 4 showing representation of latent space in case of training and testing data. |
Reviewer 2 Report
A method for identifying anomalous data from a robot is proposed in the paper. The use of Variational AutoEncoder is investigated to identify anomalous data. A Monte Carlo based technique is used to determine nominal and anomalous data. Experiments are carried out demonstrating that the technique is capable of reliable identifying anomaly. Six figures clarify the used methods.
Remarks:
- both sections 6. Discussion and 7. Conclusion are short - maybe to expand them or join together
- accronym VAE should be avoided in title
- Monte-carlo -> Monte Carlo
- Spelling should be checked, e.g. row 124: Figures 1a, 1b illustrates; 235: Figure 4. ... reconstruction of same same ...
The choice of models and methods is well reasoned. Real data provided by the robot hardware is used in the experiments. The used Variational AutoEncoder is one of the recent deep learning based generative models. It will be able to reproduce a sample that already appeared during training, while it will fail if a sample contains any kind of anomaly. Monte Carlo based technique has been applied to data. Monte Carlo methods can be used to solve any problem having a probabilistic interpretation (such as anomaly detection studied in this paper).
Author Response
Comment |
Reply |
both sections 6. Discussion and 7. Conclusion are short - maybe to expand them or join together |
Agreed, sections merged together |
acronym VAE should be avoided in title |
Agreed, title changed into “Variational AutoEncoder to identify anomalous data in robots” |
Monte-carlo -> Monte Carlo |
Agreed and fixed |
Spelling should be checked, e.g. row 124: Figures 1a, 1b illustrates; 235: Figure 4. ... reconstruction of same same ... |
Agreed and fixed |
Reviewer 3 Report
Dear Authors,
I have some comments on your article:
1. The legibility of figures 4 to 6 should be improved.
2. Literature should be checked if there are no newer items. Especially from the last 18 months.
3. At least some references from recent years should be added and the literature review should be updated.
4. Line 203-204 - with encoder’s layers dimensioned respectively [512, 256, 128, 64, 32] – one would have to formally distinguish the spelling of the number of layers from the style of citing literature.
5. As for a scientific article, it lacks equations describing dependencies, which are only described in the text descriptively.
Best regards
Author Response
Comment |
Reply |
The legibility of figures 4 to 6 should be improved. |
Agreed. Figure 4b has been completely modified. Colors of Figures 5 and 6 have been improved using standard color palette. Moreover, legend has been modified to improve readability. |
Literature should be checked if there are no newer items. Especially from the last 18 months. |
Add reference to paper from 2020 using similar technique. The total number of references published within 18 months is already 3 out of 7 scientific publications. |
At least some references from recent years should be added and the literature review should be updated. |
See comment above |
Line 203-204 - with encoder’s layers dimensioned respectively [512, 256, 128, 64, 32] – one would have to formally distinguish the spelling of the number of layers from the style of citing literature. |
Agreed and modified |
As for a scientific article, it lacks equations describing dependencies, which are only described in the text descriptively. |
It is unclear what equations could be added to give value. |
Reviewer 4 Report
Dear Authors,
Regarding the first round review of this manuscript, the reviewer has the following comments.
- As I understood in your research the residual signal is calculated by the difference between normal signal and measured signal. If its correct, how you can solve the challenge of uncertainties?
- Please add a block diagram regarding the proposed algorithm and explain step-by-step about your method there.
- In the Figures, these are based on samples or time? how many samples do you select and what is the sampling rate frequencies?
- Please highlighted the contributions.
- The results need to be improved.
Regards
Author Response
Comment |
Reply |
As I understood in your research the residual signal is calculated by the difference between normal signal and measured signal. If its correct, how you can solve the challenge of uncertainties? |
No, it is not correct. In our research we make use of trained Variational AutoEncoder to reproduce known signals. The VAE robustness allows us to cope with signal noise. A value scoring the difference between signals reconstructed by the VAE and measured signals is calculated. |
Please add a block diagram regarding the proposed algorithm and explain step-by-step about your method there. |
Agreed. Authors added a schematic representation of how our proposed method works. For cosmetic reasons it is presented in form of pseudo-code rather than block diagram. |
n the Figures, these are based on samples or time? how many samples do you select and what is the sampling rate frequencies? |
As stated in all the figures, time is represented as sample. Sampling time added in the text. |
Please highlighted the contributions. |
It is unclear the meaning of this comment. |
The results need to be improved |
Authors already stated that one of the weaknesses of this study is the lack of real anomaly data. This is due to the high reliability of the Kinova robots. |
Round 2
Reviewer 1 Report
the remarks have been addressed
Reviewer 3 Report
Dear Authors,
Thank you very much for introducing changes that have improved the quality of the article. I have no more comments.
Best regards
Reviewer 4 Report
Dear Authors,
Thank you for your response letter. Regarding the 2nd round review, the reviewer has the following comments:
- Based on your response in the first question, the VAE is used to improve the robustness against to uncertainty and external disturbance. How this technique improve the robustness? explain about it in the manuscript and highlighted please.
- Please highlighted your manuscript's contribution(s) in the introduction," clear ".
Regards,
Author Response
Comment | Reply |
Based on your response in the first question, the VAE is used to improve the robustness against to uncertainty and external disturbance. How this technique improve the robustness? explain about it in the manuscript and highlighted please. |
I think using the VAE for improved robustness is a distraction from the actual reasoning for the VAE use. This line from the manuscript explains the purpose behind using the VAE: "One of the main advantages of using a VAE instead of AutoEncoder is that having a stochastic process as part of the latent space permits to generate multiple reconstructions of the predicted signal starting from a single point in the latent space in a Monte Carlo fashion." However, we have added: "In future works we will investigate more the possibility of using statistics from multiple reconstructions of the same sample to identify anomalies or using a standard auto-encoder to simplify the process." To cover the sensible suggestion from the reviewer to consider using an AutoEncoder over a VAE. |
Please highlighted your manuscript's contribution(s) in the introduction," clear ". |
Added sentence at line 57 in which we clearly stated our main research contribution. |